analytical chemistry

eflornithine, Hantzsch reaction, spectrofluorimetry, pharmaceutical cream, biological analysis

**Author for correspondence:**
Mohamed A. Abdel-Lateef
e-mail: mohamed_abdellateef@azhar.edu.eg;
m_aldoyik@yahoo.com

This article has been edited by the Royal Society of Chemistry, including the commissioning, peer review process and editorial aspects up to the point of acceptance.

# Application of Hantzsch reaction for sensitive determination of eflornithine in cream, plasma and urine samples

## Albandary Almahri[1] and Mohamed A. Abdel-Lateef[2]

[1]General Courses Unit, Faculty of Sciences and Arts, King Khalid University, Dhahran Aljanoub, Saudi Arabia
[2]Department of Pharmaceutical Analytical Chemistry, Faculty of Pharmacy, Al-Azhar University, Assiut Branch, Assiut 71524, Egypt

MAA-L, 0000-0002-3020-4966

Eflornithine (EFN) is an anti-*Trypanosoma brucei* agent for the medication of sleeping sickness and widely distributed for the treatment of hirsutism (unwanted facial hair in women). The presented work demonstrates a comprehensive analytical approach for the spectrofluorometric determination of EFN in commercial cream samples and various biological samples. The proposed method is based on the formation of a highly yellow–green fluorescence dihydropyridine derivative after the interaction between EFN and acetylacetone/formaldehyde reagent in a slightly acidic medium. Furthermore, the optimal variables such as reagent volumes, pH of the medium, heating time, buffer volume, heating temperature and diluting solvent were carefully selected to achieve the maximum fluorescence activity. The fluorescence activity for the formed derivative was measured at $\lambda_{\text{emission}} = 477$ nm after $\lambda_{\text{excitation}} = 418$ nm. Concerning linearity, accuracy, sensitivity, precision and robustness, the presented method was validated and verified according to ICH guidelines. Moreover, the proposed work offered a selective determination for EFN in various brands of pharmaceutical cream without any interference from excipients. Eventually, the current approach was assured to be successful in the estimation of EFN in urine and plasma samples with acceptable recovery results.

## 1. Introduction

Eflornithine or α-difluoromethylornithine (EFN) acts as an ornithine decarboxylase (the key enzyme in polyamines

biosynthesis) inhibitor with specific and irreversible suppression effects [1,2]. This inhibition effect leads to an effective impairment for the cellular division [3]. Therefore, EFN was originally developed for the medication of cancer diseases but not marketed for this purpose. However, EFN is presently under clinical evaluation as a medication for different types of cancer diseases [4]. Currently, EFN cream is FDA-approved (2000) under the trade name of Vaniqa® cream for the treatment and reduces unwanted facial hair in women and widely licensed in most world countries as a potent and safe anti-hirsutism drug [5]. Also, EFN has a powerful antiprotozoal effect against *Trypanosoma brucei gambiense* parasite which causes human African trypanosomiasis, known as the sleeping sickness disease [6,7]. The intravenous doses of EFN were approved by the FDA in 1990 under the trade name of Ornidyl® for the medication of the meningoencephalic stage of human African trypanosomiasis [8]. Accordingly, EFN was licensed as a vials pharmaceutical dosage form for the medication of sleeping sickness in Europe, the USA and many countries of Africa [9,10]. Therefore, the estimation of EFN in various biological media has a particular value for investigating pharmacodynamics and pharmacokinetics as well as monitoring the absorption and tissue uptake of EFN in both pre-clinical and clinical research [11]. A survey of the literature explained that there are several analytical chromatographic methods such as LC-evaporative light scattering detector [12], LC-UV detector [13–15], LC-MS/MS detector [16] and few spectrophotometric methods (UV and colorimetric) [17–20] have been reported to estimate EFN. In addition, one enzyme-linked immunosorbent assay method [21] and one electrochemical method also have been reported [11]. However, the expensive cost, the time loss, ultrafiltration, the harmful organic solvents consumption and the sophisticated laboratory set-up are known to be the major drawbacks of the above-mentioned liquid chromatographic methods [12–16]. In addition, the reported spectrophotometric methods are limited to analyse the cited drug in vials dosage only without broaching to cream samples or the biological fluid analysis [17–20]. Hence, a cost-effective and uncomplicated analytical method with sufficient levels of sensitivity and selectivity is required for the quantitation of EFN in biological fluids and pharmaceutical cream. Analytically, the spectrofluorimetric technique is particularly used for the quantitation of the fluorescent substances in the pharmaceutical samples as well as in the biological media due to its sensitivity, simplicity, rapidity, laboratory availability, selectivity and economical features [22–24]. So, the estimation of EFN using the spectrofluorimetric method attracts great interest. EFN is a rigid molecule and not bearing any fluorophoric nucleus in its structure. Therefore, no fluorometric method had been reported for the determination of EFN. Accordingly, the current work is the first attempt to develop an inexpensive, efficient, well-validated and easily operated spectrofluorometric method for the sensitive and selective quantification of EFN in its pharmaceutical cream and biological fluids. The method in the presented work is based on the conversion of EFN to a high-fluorescence compound through the condensation interaction between its primary amino group and acetylacetone/formaldehyde as a low-cost fluorogenic reagent.

# 2. Experimental procedure

## 2.1. Materials, reagents and solvents

All the solvents, reagents and chemicals used to generate this work were in a high purity state. EFN hydrochloride, Beta-cyclodextrin, formaldehyde solution (40%) and Tween-80 were purchased from Sigma-Aldrich, USA. Hydrochloric acid, acetylacetone, phosphoric acid, sodium hydroxide, hexane, sodium dodecyl sulfate and dimethylsulfoxide were bought from Alpha Chemika, Mumbai, India. Methanol, ethanol, acetonitrile, dimethylformamide and acetone were bought from El-Nasr Cairo, Egypt. The trichloroacetic acid powder was bought from Research–LAB, Mumbai India. Flornith® cream and Eflotism® cream are products of Mash Premiere (New Cairo, Egypt) and Multi-Apex (Badr City, Egypt), respectively.

## 2.2. Apparatus

The fluorescence spectrometer (serial no: FS-1304002, Scinco, Korea) supplied with a 150 W Xe-arc lamp was used throughout this work. All measurements were performed using 5 nm slit width for both monochromators.

## 2.3. Preparation of standard, buffer and reagent solutions

The powder of EFN (20 mg) was dissolved in 100 ml of distilled water and prepared at a concentration of $200 \, \mu g \, ml^{-1}$. The solution of (Teorell and Stenhagen) buffer was prepared by mixing 100 ml of 0.51 M phosphoric acid, 100 ml of 0.33 M citric acid and 343 ml of 1.0 M sodium hydroxide; then the mixture was made up to 1000 ml with water [25]. Then, the required pH values (2.0–7.0) were achieved by adding the appropriate volume of 0.1 M hydrochloric acid solution [25]. The solution of acetylacetone reagent (8.0 ml) was mixed with methanol (92.0 ml) and prepared at the concentration of 8.0% v/v, while the solution of formaldehyde reagent (20.0 ml) was mixed with water (80.0 ml) and prepared at the final reagent concentration of 8.0% v/v.

## 2.4. General procedures for the analysis

Into a series of test tubes (consolidated in a stainless-steel rack), 0.8 ml of Teorell and Stenhagen solution (pH = 6.2), 0.6 ml of acetylacetone solution and 1.5 ml of formaldehyde solution were added and mixed with proper concentrations of EFN standard solution in the range of $1.0–8.0 \, \mu g \, ml^{-1}$. Tubes were shaken well and the rack content was placed on a boiling water bath for 20 min then cooled to $25 \pm 5°C$. Then the content was transferred to a 10 ml calibrated flask by washing the tube residue with 1 ml of methanol. Then the flask was filled to the mark with water. Solutions were scrutinized at the fluorescence emission of $\lambda_{emission} = 477 \, nm$ after the excitation of $\lambda_{excitation} = 418 \, nm$.

## 2.5. Pharmaceutical cream samples (extraction and assay)

A proper amount of Flornith® or Eflotism® cream that is equal to 100 mg of EFN hydrochloride was weighed and separately transferred into a 100 ml volumetric flask; then about 40 ml of hot distilled water was added and the content was sonicated for 15 min and cooled to $20 \pm 5°C$, and filled up to 100 ml with water and shaken well. Then after the content was poured into 250 ml separating funnel and gently shaken numerous times with about 30 ml hexane then the content was cooled in the refrigerator at the temperature of 5°C to separate and exclude the fatty content from samples. The separated aqueous layer was diluted with water and underwent the general procedures for analysis.

## 2.6. Steps for spiked plasma and urine samples

Into a series of tubes with a size of 10 ml, 1000 μl of drug-free human plasma was added and spiked with 100 μl of EFN hydrochloride (with the desired concentrations), then 500 μl of trichloroacetic (20%) was added to each sample. Samples were vortexed for 1 min. Then samples were placed at 4°C for 60 min to achieve a more efficacious protein precipitation process [12]. Samples were centrifuged at 10 000 r.p.m. for 10 min. Finally, the liquid phase was transferred, separated, diluted with water and underwent the analysis by the general procedures. Similarly, 1000 μl of freshly pooled drug-free human urine was added and spiked with 100 μl of EFN hydrochloride (with the desired concentration), then 500 μl of trichloroacetic (20%) was added to each sample and the samples were vortexed for 1 min. Then, the samples were placed at 4°C for 60 min. Samples were centrifuged at 10 000 r.p.m. for 10 min. Finally, the liquid phase was transferred, separated, diluted with water and underwent the analysis by the general procedures.

# 3. Results and discussion

Despite eflornithine hydrochloride molecule does not having fluorescence properties, it can be converted into a highly fluorescent molecule through the derivatization of its amino group by a fluorogenic agent. Recently, the Hantzsch reaction was applied for the spectrofluorometric determination of a wide range of the non-fluorescent active ingredients (containing primary amino group) either in their dosage form or various biological samples through the formation of highly fluorescent dihydropyridine derivatives in the slightly acidic medium [26–28]. Therefore, the condensation reaction between acetylacetone, formaldehyde and EFN results in a turn-on fluorescence activity of EFN; accordingly, the studied drug can be quantitatively measured through the spectrofluorometric technique (scheme 1). The formed dihydropyridine product exhibited a fluorescence activity at $\lambda_{emission} = 477 \, nm$ upon its excitation at $\lambda_{excitation} = 418 \, nm$ (figure 1).

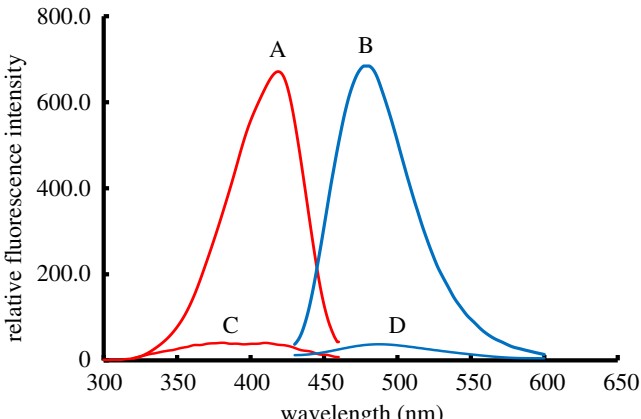

**Scheme 1.** The suggested reaction mechanism for the interaction between EFN and acetylacetone/formaldehyde in a slightly acidic medium.

**Figure 1.** Fluorescence spectra (C, D) for the blank and (A, B) the reaction product of EFN (500.0 ng ml$^{-1}$) with acetylacetone/formaldehyde at pH = 6.2.

## 3.1. Optimization of fluorescence turn-on conditions

The interaction between EFN, acetylacetone and formaldehyde is highly affected by the variation in reaction conditions such as heating temperature, heating time, the volume of acetylacetone, volume of buffer, pH value, volume of formaldehyde and diluting solvent. Therefore, the influence of each factor on the formation of dihydropyridine derivative was separately examined and optimized.

### 3.1.1. Acetylacetone and formaldehyde volume

The appropriate volumes for both acetylacetone solution (8.0% v/v) and formaldehyde solution (8.0% v/v) were carefully examined at the volume range of 0.1–2.0 ml. It was noted that the best fluorescence strength for the formed dihydropyridine derivative was obtained in the volume range of 0.4–0.8 ml from acetylacetone solution, higher or lower than this range leads to a diminishing fluorescence activity (figure 2). While the fluorescence intensity was gradually augmented with the increase in the volume of formaldehyde up to 1.2 ml and remained without significant variation above this volume (figure 2).

### 3.1.2. Heating temperature and time

Varied heating temperatures were tested in this study (80°C, 90°C and water boiling point). It was noted that the heating temperature is an important factor for the formation of the product and the best fluorescence intensity was obtained after heating the content at the boiling point (figure 3). Besides that, the best fluorescence turn-on results were obtained after heating the tubes for 35 min. Hence, we heat the content at 100°C for 40 min to achieve the highest fluorescence turn-on value (figure 3).

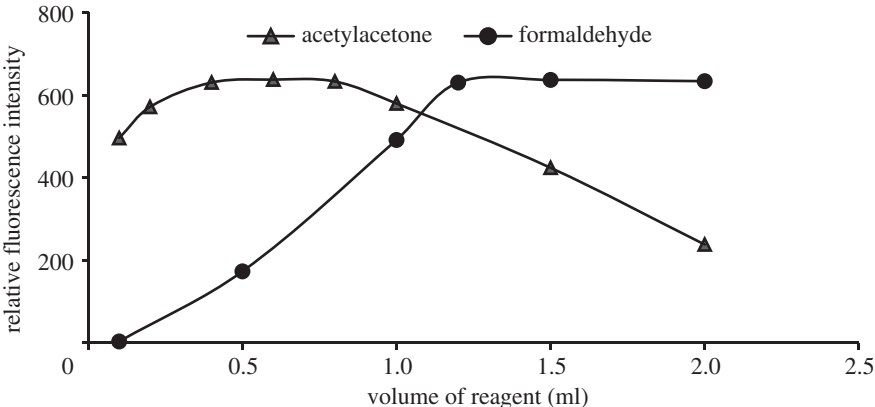

**Figure 2.** Effect of reagents volumes on the formation of the reaction product between EFN (500.0 ng ml$^{-1}$) and acetylacetone/formaldehyde (8.0% v/v).

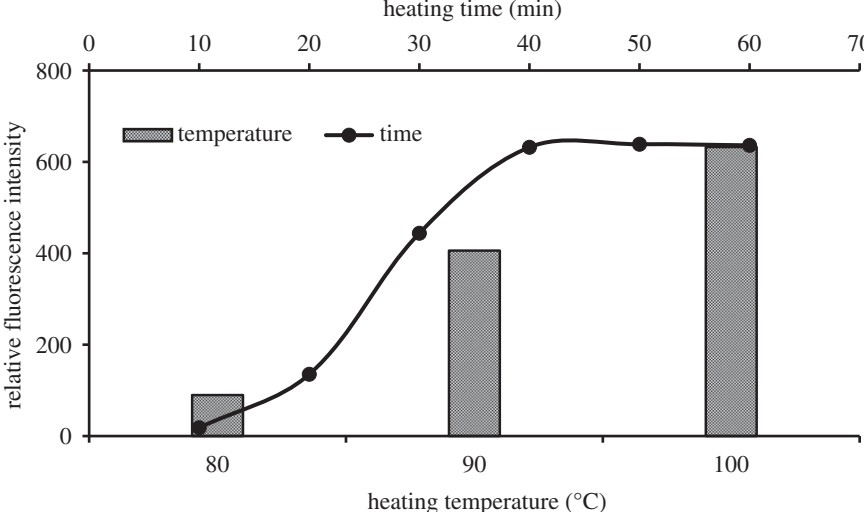

**Figure 3.** Effect of heating temperature and heating time on the formation of the reaction product of between EFN (500.0 ng ml$^{-1}$) and acetylacetone/formaldehyde (8.0% v/v).

### 3.1.3. pH value and buffer volume

Varied pH values were tested in the range of 2.0–7.0. It was noted that the formation of the reaction is highly affected by the variation in pH value, where the lowest fluorescence turn-on results were obtained in the pH range of 2–5 and the best fluorescence turn-on results were achieved at the pH range of 5.8–6.8 (figure 4). Therefore, the pH value of 6.2 is the appropriate value to obtain the maximum fluorescence turn-on activity. Besides that, the proper amount from Teorell buffer was also tested in the volume range of 0.1–2.0 ml. It was also found that the fluorescence strength was gradually augmented with the increase in the volume up to 0.6 ml, and remained without significant variation after the increase in the Teorell buffer volume (figure 4).

### 3.1.4. Diluting solvents and micellar media

A collection from the common solvents was tested as dilutant to examine which one produces the best fluorescence turn-on activity. The tested solvents were dimethylsulfoxide, methanol, acetic acid in methanol (0.1 M), acetonitrile, water, ethanol, phosphoric acid in ethanol (0.1 M), dimethylformamide and acetone. Besides that, various classes of surfactants included cationic surfactant (hexadecyl trimethyl ammonium bromide), non-ionic surfactant (Tween-80) and anionic surfactant (sodium dodecyl sulfate) and β-cyclodextrin were also tested in the aqueous medium to investigate their enhancement effect on the fluorescence activity of the reaction product. It was noted that the best result for the fluorescence turn-on was achieved with water and the lowest results were achieved with

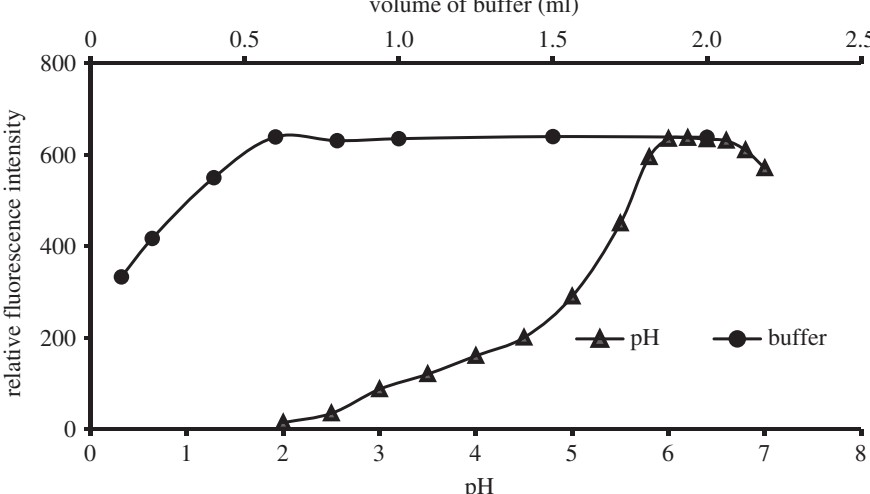

**Figure 4.** Effect of pH value and buffer volume on the formation of the reaction product between EFN (500.0 ng ml$^{-1}$) and acetylacetone/formaldehyde (8.0% v/v).

dimethylsulfoxide and dimethylformamide solutions. Besides that, the fluorescence activity is not affected by the addition of any class of surfactants or macromolecule. Hence, the proper diluting solvent was water.

## 3.2. Fluorescence quantum yield ($\Phi$)

The degree of fluorescence turn-on for EFN (fluorophore brightness) can be inferred from a quantum yield value of the formed dihydropyridine derivative and can be determined by the equation [29]

$$\Phi = \Phi_{\text{fluorescein}} \times \frac{F_{\text{reaction product}}}{A_{\text{reaction product}}} \times \frac{A_{\text{fluorescein}}}{F_{\text{fluorescein}}},$$

where $\Phi$ is referring to the produced quantum yield for the reaction product, $\Phi_{\text{fluorescein}}$ is referring to the quantum yield for fluorescein, $F$ is referring to integrated fluorescence intensity for the reaction product and fluorescein and $A$ is referring to the absorbance for the reaction product and fluorescein. It was found that the $\Phi$ value for the reaction product was 38.2%.

## 3.3. Validation

The validity for the presented method was achieved through verifying parameters of accuracy, selectivity, sensitivity, robustness, linear range and precision depending on ICH guidelines [30].

### 3.3.1. Linearity and range [30]

At the optimum interaction conditions, the linearity between the concentration of EFN and the fluorescence intensity of the reaction product was shown in the range of 100.0–800.0 ng ml$^{-1}$ with 0.9996 for the correlation coefficient value. The regression equation and the calculated analytical parameters are offered in table 1.

### 3.3.2. Sensitivity

The detection limit (LOD) and the quantitation limit (LOQ) values for the current work were calculated for the concentration of EFN by the adoption of the equations [30]

$$\text{LOQ} = \frac{10 \times S_a}{b}$$

**Table 1.** Analytical parameters for the method.

| parameter | fluorometric assay |
|---|---|
| linear range (ng ml$^{-1}$) | 100.0–800.0 |
| slope | 1.17 |
| s.d. of slope | 0.0174 |
| intercept | 55.82 |
| s.d. of intercept | 7.07 |
| s.d. of residuals ($S_{y/x}$) | 8.28 |
| correlation coefficients ($r$) | 0.9996 |
| determination coefficients ($r^2$) | 0.9993 |
| number of determinations | 6 |
| LOQ (ng ml$^{-1}$) | 60.67 |
| LOD (ng ml$^{-1}$) | 20.02 |

**Table 2.** The comparison between the proposed method and the reported spectroscopic methods.

| method | reagent | linear range (µg ml$^{-1}$) | LOQ | LOD | sample | ref. |
|---|---|---|---|---|---|---|
| photometric | dansyl chloride | 0.5–4.5 | 0.052 | 0.017 | bulk | [17] |
| photometric | vanillin | 5–25 | 4.3 | 1.2 | vials | [18] |
| photometric | 1,2-naphthoquinone-4-sulfonate | 10–70 | 0.52 | 0.172 | bulk | [19] |
| photometric | *para*-dimethylamino benzaldehyde | 5–40 | 3.94 | 1.184 | vials | [20] |
| proposed | acetylacetone/formaldehyde | 0.1–0.8 | 0.06 | 0.02 | cream, urine and plasma | |

and

$$\text{LOD} = \frac{3.3 \times S_a}{b},$$

where $b$ and $S_a$ are values for slope and s.d. of the intercept, respectively. The calculated values were 60.67 and 20.02 µg ml$^{-1}$ for LOQ and LOD, respectively. The comparison between the linear range and the sensitivity for the proposed method and the reported spectroscopic methods is offered in table 2.

### 3.3.3. Precision, robustness and accuracy [30]

The validation was performed by determining EFN in its pure form using three varied concentrations of EFN within the mentioned linear range (triplicate measurements for each concentration levels). RSD values in table 3 refer to the acceptable precision for the suggested method. In a similar way, the accuracy parameter for the current method was concluded and accepted from the obtained percentage recovery values in table 3. Lastly, the robustness parameter was assessed against the minor variations in the optimum reaction conditions such as the volume of acetylacetone (0.6 ± 0.1 ml), the value of pH (6.2 ± 0.1), volume of formaldehyde (1.5 ± 0.1 ml) and heating time (40 ± 3 min). Data in table 4 refer to the robustness of the current method against any minor changes.

**Table 3.** Precision and accuracy data for the determination of EFN by the current method.

| validation parameter | ng ml$^{-1}$ | % recovery $\pm$ RSD[a] |
|---|---|---|
| inter-day precision | 200.0 | 98.78 $\pm$ 1.75 |
| | 400.0 | 99.32 $\pm$ 1.66 |
| | 600.0 | 100.46 $\pm$ 1.89 |
| intra-day precision | 200.0 | 99.93 $\pm$ 1.63 |
| | 400.0 | 100.25 $\pm$ 1.50 |
| | 600.0 | 101.70 $\pm$ 1.2 |
| | ng ml$^{-1}$ | % recovery $\pm$ s.d.[a] |
| accuracy | 200.0 | 101.08 $\pm$ 1.94 |
| | 500.0 | 98.42 $\pm$ 1.32 |
| | 700.0 | 100.15 $\pm$ 1.67 |

[a]s.d. and RSD are standard deviation value and relative standard deviation value, respectively.

**Table 4.** Robustness data for the determination of EFN by the current method.

| reaction variable | % recovery $\pm$ s.d.[a] |
|---|---|
| pH | |
| 6.1 | 98.31 $\pm$ 1.03 |
| 6.3 | 101.70 $\pm$ 1.39 |
| volume of Teorell buffer | |
| 0.7 ml | 99.69 $\pm$ 1.64 |
| 0.9 ml | 101.18 $\pm$ 1.32 |
| heating time | |
| 43 min | 100.32 $\pm$ 1.81 |
| 37 min | 100.38 $\pm$ 0.96 |
| volume of acetylacetone | |
| 0.5 ml | 98.65 $\pm$ 1.21 |
| 0.7 ml | 101.76 $\pm$ 1.03 |
| volume of formaldehyde | |
| 1.4 ml | 99.11 $\pm$ 1.27 |
| 1.6 ml | 98.08 $\pm$ 1.66 |

[a]EFN conc. 500.0 ng ml$^{-1}$.

# 4. Applications

## 4.1. Pharmaceutical cream samples

The current method was practically applied to estimate the content of EFN in two commercial brands of the pharmaceutical cream (Eflotism® cream and Flornith® cream). Results were statistically compared with the reported method [17]. The precision and accuracy of the current method were ascertained through the application of Student's *t*-test and variance ratio *F*-test, using statistical analysis. The statistical analysis in table 5 is referred to that the obtained results from both methods are in an acceptable agreement.

**Table 5.** Estimation of EFN in varied brands of cream samples by the current method.

| | % recovery[a] ± s.d. | | | |
|---|---|---|---|---|
| dosage form | proposed | reported | $t$-value[b] | $F$-value[b] |
| Eflotism® 15 g cream | 100.79 ± 1.47 | 99.24 ± 1.51 | 1.63 | 0.94 |
| Flornith® 30 g cream | 101.07 ± 2.17 | 100.68 ± 2.43 | 0.26 | 1.25 |

[a]Average of five determinations.
[b]Tabulated value at 95% confidence limit; $F = 6.338$ and $t = 2.306$.

## 4.2. Application on biological samples

After the parental (intravenous) administration of EFN, the maximal concentration ($C_{max} = 10.03\ \mu g\ ml^{-1}$) values are achieved after about 4 h, with the half-life ($t_{1/2}$) of 3 h and 80% of EFN dose was excreted as an unaltered drug in the urine without any biotransformation [31,32]. Accordingly, it is possible to analyse the aforementioned drug in plasma samples and urine samples using the current method. The final percentage recovery results for the estimation of EFN in spiked plasma samples were 94.57 ± 2.63, 103.15 ± 2.7 and 100.07 ± 2.56, while the calculated percentage recovery results for the estimation of EFN in spiked urine samples were 102.63 ± 1.81, 97.98 ± 1.97 and 98.11 ± 2.07.

## 5. Conclusion

The current work is the first fluorometric method to generate inexpensive, efficient, well-validated and easily operated spectrofluorometric procedures for the sensitive and selective quantification of EFN in pharmaceutical cream and different biological fluids (urine and plasma). The current method offers substantial merits over the reported chromatographic methods such as ignoring the ultrafiltration process, eliminating the sophisticated procedures and excluding the harmful organic solvents. In addition, the current method offers the options of saving money, time and effort for the operator. Furthermore, the work is distinguished by sensitivity, laboratory availability and simplicity. Therefore, the current method can be used as an alternative to the HPLC methods for the analysis of EFN by industrial (quality control) and research laboratories.

Data accessibility. Data available from the Dryad Digital Repository: https://doi.org/10.5061/dryad.zs7h44j7x [33].
Authors' contributions. M.A.A.-L. carried out the laboratory work, participated in data analysis, participated in the design of the study and submitted the manuscript. A.A. designed the study, coordinated the study, participated in data analysis, conceived the study and drafted the manuscript. All authors gave final approval for publication.
Competing interests. We declare we have no competing interests.
Funding. We received no funding for this study.
Acknowledgement. The authors would like to express their gratitude to King Khalid University, Saudi Arabia for providing administrative and technical support.

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
