## [Peer Review File · Royal Society Open Science]

Review History

RSOS-210366.R0 (Original submission)

Review form: Reviewer 1

Is the manuscript scientifically sound in its present form?

Yes

Are the interpretations and conclusions justified by the results?

Yes

Is the language acceptable?

Yes

Do you have any ethical concerns with this paper?

No

Have you any concerns about statistical analyses in this paper?

No

Recommendation?

Accept with minor revision (please list in comments)

Comments to the Author(s)

Minor corrections

- 1- Page No 2, the word "Preciseness" corrects to "precision".
- 2- Page No 2, the sentence "varied biological media" rewrite to be "various biological samples"
- 3- The word "Specimens" should be correct as "samples" in the whole manuscript.
- 4- Page No 4, write "method" in place of "methodology" in the whole manuscript
- 5- The word "formed product" change to "reaction product" in the whole manuscript.
- 6- Page No 9, "The validity for the "work". Change the word "work" by "method".
- 7- Page No 9, "...ICH guiding principles" correct to "..... ICH guidelines"
- 8- Page No 9, linear range correct as "linearity and range".
- 9- Page No 8, "various classes surfactants (cationic, nonionic and anionic)". Please mentioned names of the utilized surfactants.
- 10- Scheme 1, pleas mention the reaction conditions such as pH of the solution.

Review form: Reviewer 2

Is the manuscript scientifically sound in its present form?

Yes

Are the interpretations and conclusions justified by the results?

Yes

Is the language acceptable?

Yes

Do you have any ethical concerns with this paper?

No

Have you any concerns about statistical analyses in this paper?

Yes

Recommendation?

Accept with minor revision (please list in comments)

Comments to the Author(s)

In this manuscript Almahri and co-workers developed a method based on Hantzsch reaction and applied for the determination of eflornithine in cream, plasma and urine samples. It was well designed and clear. After minor corrections, it can be acceptable.

1. Spectrofluorometric conditions such as light source, excitation and emission slits, etc. must be given.
2. Figure 1 should be corrected. All graphs are line form.
3. LOD and LOQ values can be given in Table 1.
4. It is better to compare the proposed method with previous method in field of LOD-LOQ values, linearity, recovery and etc.
5. Some typos are in the text, they should be corrected.

Decision letter (RSOS-210366.R0)

Dear Dr Abdel-Lateef:

Title: Application of Hantzsch reaction for sensitive determination of eflornithine in cream, plasma and urine samples
Manuscript ID: RSOS-210366

Thank you for submitting the above manuscript to Royal Society Open Science. On behalf of the Editors and the Royal Society of Chemistry, I am pleased to inform you that your manuscript will be accepted for publication in Royal Society Open Science subject to minor revision in accordance with the referee suggestions. Please find the reviewers' comments at the end of this email.

The reviewers and handling editors have recommended publication, but also suggest some minor revisions to your manuscript. Therefore, I invite you to respond to the comments and revise your manuscript.

Because the schedule for publication is very tight, it is a condition of publication that you submit the revised version of your manuscript before 07-Apr-2021. Please note that the revision deadline will expire at 00.00am on this date. If you do not think you will be able to meet this date please let me know immediately.

- 1) A text file of the manuscript (tex, txt, rtf, docx or doc), references, tables (including captions) and figure captions. Do not upload a PDF as your "Main Document".
- 2) A separate electronic file of each figure (EPS or print-quality PDF preferred (either format should be produced directly from original creation package), or original software format)
- 3) Included a 100 word media summary of your paper when requested at submission. Please ensure you have entered correct contact details (email, institution and telephone) in your user account
- 4) Included the raw data to support the claims made in your paper. You can either include your data as electronic supplementary material or upload to a repository and include the relevant doi within your manuscript
- 5) All supplementary materials accompanying an accepted article will be treated as in their final form. Note that the Royal Society will neither edit nor typeset supplementary material and it will

be hosted as provided. Please ensure that the supplementary material includes the paper details where possible (authors, article title, journal name).

Kind regards,
Dr Laura Smith
Publishing Editor, Journals

RSC Associate Editor:
Comments to the Author:
(There are no comments.)

RSC Associate Editor:
Comments to the Author:
(There are no comments.)

Reviewer comments to Author:
Reviewer: 1

Comments to the Author(s)
Minor corrections

- 1- Page No 2, the word "Preciseness" corrects to "precision".
- 2- Page No 2, the sentence "varied biological media" rewrite to be "various biological samples"
- 3- The word "Specimens" should be correct as "samples" in the whole manuscript.
- 4- Page No 4, write "method" in place of "methodology" in the whole manuscript
- 5- The word "formed product" change to "reaction product" in the whole manuscript.
- 6- Page No 9, "The validity for the "work". Change the word "work" by "method".
- 7- Page No 9, "...ICH guiding principles" correct to "..... ICH guidelines"
- 8- Page No 9, linear range correct as "linearity and range".

- 9- Page No 8, “various classes surfactants (cationic, nonionic and anionic)”. Please mentioned names of the utilized surfactants.
- 10- Scheme 1, pleas mention the reaction conditions such as pH of the solution.

Reviewer: 2

Comments to the Author(s)

In this manuscript Almahri and co-workers developed a method based on Hantzsch reaction and applied for the determination of eflornithine in cream, plasma and urine samples. It was well designed and clear. After minor corrections, it can be acceptable.

1. Spectrofluorometric conditions such as light source, excitation and emission slits, etc. must be given.
2. Figure 1 should be corrected. All graphs are line form.
3. LOD and LOQ values can be given in Table 1.
4. It is better to compare the proposed method with previous method in field of LOD-LOQ values, linearity, recovery and etc.
5. Some typos are in the text, they should be corrected.

Author's Response to Decision Letter for (RSOS-210366.R0)

See Appendix A.

Decision letter (RSOS-210366.R1)

Dear Dr Abdel-Lateef:

Title: Application of Hantzsch reaction for sensitive determination of eflornithine in cream, plasma and urine samples
Manuscript ID: RSOS-210366.R1

It is a pleasure to accept your manuscript in its current form for publication in Royal Society Open Science. The chemistry content of Royal Society Open Science is published in collaboration with the Royal Society of Chemistry.

RSC Associate Editor
Comments to the Author:
(There are no comments.)

Reviewer(s)' Comments to Author:

Appendix A

Dear Publishing Editor, Dr Laura Smith,

Dear Subject Editor Professor Anthony Stace,

Dear Associate Editor Dr Ya-Wen Wang

On behalf of all authors, I would like to thank you and all the editorial team members for the opportunity that we have been given to further revise our manuscript **ID RSOS-210366 entitled "Application of Hantzsch reaction for sensitive determination of eflornithine in cream, plasma and urine samples"** for its publication in Royal Society Open Science. We have carefully revised our manuscript according to the reviewer comments. We are grateful for the comments and advice that we have received. Any further questions about this matter are greatly welcomed.

N.B. All corrections have been re-written in red color in the revised manuscript.

Response to reviewer #1:

Comments to the Author(s)

Minor corrections

1-“Page No 2, the word “Preciseness” corrects to “precision”.

The word was corrected.

2-“Page No 2, the sentence “varied biological media” rewrite to be “various biological samples”

The sentence was rewritten and highlighted with red color.

3- “The word “Specimens” should be correct as “samples” in the whole manuscript.”

The word “Specimens” was changed to “samples” in the whole manuscript and highlighted with red color.

4- "Page No 4, write "method" in place of "methodology" in the whole manuscript".

The "methodology" was changed to "method" in the whole manuscript.

5- "The word "formed product" change to "reaction product" in the whole manuscript."

Done.

6- Page No 9, "The validity for the "work". Change the word "work" by "method".

Done.

7- Page No 9, "...ICH guiding principles" correct to "..... ICH guidelines"

Done.

8- "Page No 9, linear range correct as "linearity and range".

Done

9- "Page No 8, "various classes surfactants (cationic, nonionic and anionic)". Please mentioned names of the utilized surfactants."

The utilized surfactants were mentioned in the text and highlighted with red color.

10- Scheme 1, pleas mention the reaction conditions such as pH of the solution.

Done.

Response to reviewer #2:

Comments to the Author(s)

In this manuscript Almahri and co-workers developed a method based on Hantzsch reaction and applied for the determination of eflornithine in cream,

plasma and urine samples. It was well designed and clear. After minor corrections, it can be acceptable.

1. "Spectrofluorometric conditions such as light source, excitation and emission slits, etc. must be given."

The required data were added under the "2. Experimental" section and highlighted with red color.

2. "Figure 1 should be corrected. All graphs are line form."

Done

3. LOD and LOQ values can be given in Table 1.

Values were added in Table 1.

4. It is better to compare the proposed method with previous method in field of LOD-LOQ values, linearity, recovery and etc.

The required comparison was prepared in Table 2.

5. Some typos are in the text, they should be corrected.

Typos were corrected.

Sincerely yours

Mohamed Abdel-Lateef, Ph.D.